# Characterization of Molecular Chaperone GroEL as a Potential Virulence Factor in *Cronobacter sakazakii*

**DOI:** 10.3390/foods12183404

**Published:** 2023-09-12

**Authors:** Dongdong Zhu, Yufei Fan, Xiaoyi Wang, Ping Li, Yaping Huang, Jingbo Jiao, Chumin Zhao, Yue Li, Shuo Wang, Xinjun Du

**Affiliations:** 1State Key Laboratory of Food Nutrition and Safety, College of Food Science and Engineering, Tianjin University of Science and Technology, Tianjin 300457, China; duiduizdd@mail.tust.edu.cn (D.Z.); 17908002@mail.tust.edu.cn (Y.F.); rebeccamargaret@msn.cn (X.W.); zoelxx@tust.edu.cn (P.L.); huangyp555@163.com (Y.H.); 18753391899@163.com (J.J.); zcm3278@163.com (C.Z.); i_liyue2020@163.com (Y.L.); s.wang@tust.edu.cn (S.W.); 2Tianjin Key Laboratory of Food Science and Health, School of Medicine, Nankai University, Tianjin 300071, China

**Keywords:** *Cronobacter sakazakii*, GroEL, export, virulent effects

## Abstract

The molecular chaperone GroEL of *C. sakazakii*, a highly conserved protein encoded by the gene *grol*, has the basic function of responding to heat shock, thus enhancing the bacterium’s adaptation to dry and high-temperature environments, which poses a threat to food safety and human health. Our previous study demonstrated that GroEL was found in the bacterial membrane fraction and caused a strong immune response in *C. sakazakii*. In this study, we tried to elucidate the subcellular location and virulent effects of GroEL. In live *C. sakazakii* cells, GroEL existed in both the soluble and insoluble fractions. To study the secretory mechanism of GroEL protein, a non-reduced Western immunoblot was used to analyze the form of the protein, and the result showed that the exported GroEL protein was mainly in monomeric form. The exported GroEL could also be located on bacterial surface. To further research the virulent effect of *C. sakazakii* GroEL, an indirect immunofluorescence assay was used to detect the adhesion of recombinant GroEL protein to HCT-8 cells. The results indicated that the recombinant GroEL protein could adhere to HCT-8 cells in a short period of time. The recombinant GroEL protein could activate the NF-κB signaling pathway to release more pro-inflammatory cytokines (TNF-α, IL-6 and IL-8), downregulating the expression of tight-junction proteins (claudin-1, occluding, ZO-1 and ZO-2), which collectively resulted in dose-dependent virulent effects on host cells. Inhibition of the *grol* gene expression resulted in a significant decrease in bacterial adhesion to and invasion of HCT-8 cells. Moreover, the deficient GroEL also caused slow growth, decreased biofilm formation, defective motility and abnormal filamentation of the bacteria. In brief, *C. sakazakii* GroEL was an important virulence factor. This protein was not only crucial for the physiological activity of *C. sakazakii* but could also be secreted to enhance the bacterium’s adhesion and invasion capabilities.

## 1. Introduction

*Cronobacter sakazakii*, an opportunistic Gram-negative pathogen mainly found to be associated with milk and cheese protein powders, was reported to cause both intestinal and systemic human diseases, such as meningitis, bacteremia and necrotizing enterocolitis through complex bacterium–host interactions [1]. Studies have shown that its main infection targets are neonates, infants and elderly individuals [2]. In 2010, two mexican infants were infected with *C. sakazakii* and developed bloody diarrhea [3]. In 2016, a premature female infant was diagnosed with sepsis caused by *C. sakazakii* ultimately leading to cerebral liquefaction and necrosis [4]. *C. sakazakii* is not thermotolerant (survives between 6 and 45 °C) but can proliferate in dry stress, even at a low water activity level of 0.3 [5]. It was reported that 18.75% of quick-frozen food collected from 39 cities in china were contaminated with *C. sakazakii* in 576 samples, wherein the contamination of frozen flour products was up to 44.34% [6]. In addition, opened powdered formula and breast pump parts were highly susceptible to contamination with *C. sakazakii* [7].

To date, many studies have been performed to investigate the pathogenic mechanism of this bacterium, and the usual virulence factors in Gram-negative bacteria, including lipopolysaccharide (LPS), enterotoxin, flagella and some outer membrane proteins, were reported to be involved in the pathogenicity of *C. sakazakii* [8,9,10,11]. Other virulence-associated factors of *C. sakazakii*, such as the bcsABC operon controlling cellulose synthesis, methyl-accepting chemotaxis protein Mcp and UDP-N-acetylglucosamine acyltransferase’s ligand Labp, are also reported to contribute to the pathogenicity of the bacterium [12,13,14]. However, although these virulence-associated factors are shown to be associated with *C. sakazakii* infection, the key virulence determinants of *C. sakazakii* and the details of this pathogen’s infection of host cells are largely unknown. Recently, many microbiological secretome studies have shown the release of various cytoplasmic proteins into the medium supernatant, especially evolutionarily conserved proteins, including glycolytic enzymes, translation factors and chaperones [15]. The secretory protein was crucial for bacteria to obtain nourishment, adapt to the environment and infect the host. Whether these exported proteins of cytoplasm are caused by accidental bacterial lysis or secreted through a special pathway remains hotly debated [16,17,18], while studies have shown that these proteins can adhere to the surface of microorganisms and have immunogenicity [19]. The virulence-associated roles of these proteins require more detailed studies.

In our previous work, the molecular chaperone GroEL was shown to be expressed at a high level in *C. sakazakii* and to cause a strong immune response [20]. The basic function of GroEL is to participate in the regulation of the heat shock response to maintain cellular homeostasis and form a nanocage structure with the co-chaperonin protein GroES to rescue the proteins from inappropriate folding and aggregation [21]. GroEL mutation was reported to cause the accumulation of a large number of newly translated peptides [22]. In addition, GroEL is also a familiar member in the secretome of eukaryotes and prokaryotes. Studies have shown that the exported GroEL of different prokaryotes can adhere to different human cells [23,24], and GroEL can produce immune protection in immune response and reduce the degree of lesions and the mortality of related diseases to a certain extent [25]. Although GroEL can act as an adhesin to aggravate specific tissue diseases caused by several corresponding prokaryotes, whether it is an important virulence factor in related diseases caused by *C. sakazakii* is still unknown.

Therefore, the purpose of this study was to identify whether *C. sakazakii* can export GroEL and whether GroEL serves as a virulence factor of this bacterium. This study indicated that *C. sakazakii* can export monomeric GroEL. The GroEL protein was able to adhere to the intestinal epithelium cells and caused adverse effects. Inhibition of the *grol* gene expression not only caused physical defects in *C. sakazakii* but also reduce the bacterial adhesion to and invasion of hosts. In brief, we proved that *C. sakazakii* GroEL is an important virulence factor, and this study will provide new insight into the pathogenicity of *C. sakazakii*.

## 2. Materials and Methods

### 2.1. Bacterial Strains and Growth Conditions

In this study, *Escherichia coli* BL21 (DE3) and *C. sakazakii* ATCC29544 strains stored at the Tianjin University of Science and Technology were used. Both bacterial strains were grown in Luria–Bertani (LB) medium at 37 °C under constant shaking unless otherwise stated.

### 2.2. Bacterial Fractionation

The bacterial fractionation was performed using Goulhen’s method with some modifications [26]. Briefly, *C. sakazakii* was cultured to an OD_600_ of 1.0 (1 × 10^10^ CFU/mL), followed by centrifugation to separate the medium supernatant from the bacterial cells. The crude outer membrane vesicle (OMV) was harvested using the saturated ammonium sulfate precipitation method from the medium supernatant and then ultracentrifuged at 250,000× *g* for 1 h at 4 °C using a supercentrifuge (Beckman, Bria, CA, USA) to obtain the purified OMV. The ultracentrifuged supernatant was rich in other secreted proteins. The periplasmic fractions were isolated through the osmotic shock method. The residual pellet was broken through mild sonication and then centrifuged to separate the crude membrane envelope and the cytoplasmic fractions. The crude membrane envelope was re-suspended in 2% Triton X-100 containing 10 mM MgCl_2_ after washing three times with TBS buffer. The suspension was ultracentrifuged at 200,000× *g* for 2 h to separate the cytoplasmic-membrane-rich cell envelope (supernatant) and the outer-membrane-rich cell envelope (pellet).

### 2.3. Identification of GroEL and Its Forms in Different Bacterial Fractions

The different bacterial fractions were separated through SDS-PAGE (12% acrylamide), and then, the protein samples were transferred to a polyvinylidene fluoride membrane (0.22 μm, 200 mA, 90 min). The membrane was blocked with 10% skim milk for 3 h in a 50 mM TBS buffer, followed by incubation with the primary antibody (anti-GroEL monoclonal antibody, Abcam, Cambridge, UK) for 2 h at room temperature. After washing three times with TBS buffer containing 0.1% Tween 20, the membrane was subsequently incubated with the horseradish-peroxidase-linked secondary antibody. HRP substrate (Millipore, Boston, MA, USA) was added, and the result was detected using an enhanced chemiluminescence detection system. As for the non-reduced Western immunoblot, we adopted Qamra’s method but with some modifications [27]. Briefly, the protein samples were separated using SDS-PAGE (7% acrylamide) and then transferred to a nitrocellulose membrane (0.45 μm). Electrophoresis was carried out at 18 V for 18 h. The following procedures of the non-reduced Western immunoblot were the same as those described above.

### 2.4. Determination of the Bacterial Surface GroEL

Detection of the GroEL localizing on the *C. sakazakii* surface was carried out using an indirect immunofluorescence assay (IIF). In brief, the tetracycline gene of the pACYC-184 plasmid was completely replaced with the *grol* gene containing the His tag using a ClonExpress^®^ II One Step Cloning Kit (Vazyme, Nanjing, China), and then, the recombinant plasmid was transformed into *C. sakazakii*. The recombinant bacteria were cultured to OD_600_ 1.0, and subsequently, the bacteria were washed three times with a PBS buffer. The bacteria were treated with an anti-His-Tag monoclonal antibody (ABclonal, Wuhan, China) (60 min incubation, 1:100 in PBS) and subsequently with the secondary antibody goat anti-mouse IgG (H+L)-FITC (Sungene Biotech, Shanghai, China) (60 min incubation, 1:2000 in PBS). The fluorescence signals were detected using a confocal microscope (Lecia Tcs Sp8, Weztlar, Germany). *C. sakazakii* containing an empty pACYC-184 plasmid was used as control. The primers for amplifying the *grol* gene (F1, R1) and the pACYC-184 backbone vector (F2, R2) are shown in Appendix A.

### 2.5. Adhesive Ability of GroEL to HCT-8 Cells

*E. coli* BL21 (DE3) containing recombinant pET-26b-*grol* was used to express recombinant GroEL (rGroEL), followed by the purification of rGroEL as described previously [28]. IIF was used to determine the adhesion of GroEL to HCT-8 cells. Briefly, the human enterocyte-like epithelial HCT-8 cells (ATCC) were maintained in RPMI 1640 medium (Gibco, Stockrick, CA, USA) containing 10% fetal bovine serum (Gibco, Stockrick, CA, USA) and then transferred to a coverglass-bottom dish for further culture until monolayer cells covered the dish bottom. After discarding the culture medium, the cells were washed three times with phosphate-buffered saline (PBS) (in each step of the subsequent treatment, the cells were washed in the same way), followed by fixing the cells with ice-cold 4% formaldehyde for 15 min and incubation with purified rGroEL at 37 °C for 30 min in serum-free RPMI 1640 medium. Subsequently, the cells were incubated with an anti-His-Tag monoclonal antibody (60 min incubation, 1:100 in PBS) and then with the secondary antibody goat anti-mouse IgG (H+L)-FITC (60 min incubation, 1:2000 in PBS). The cell nuclei were stained using 4, 6-diamidino-2-phenylindole (DAPI). The control group was prepared by using the eluent from the His-Tag Purification Resin, which was loaded with total proteins of *E. coli* BL21 (DE3) containing empty plasmid pET-26b (isoconcentration imidazole eluent) to replace rGroEL, and the other steps were the same as described above. Fluorescence was detected with a confocal microscope.

### 2.6. Determination of GroEL Protein Virulent Effects

The virulent effects of GroEL were evaluated by determining the viability of cells as described previously with some modifications [26]. Briefly, HCT-8 cells were plated on 96-well plates to yield monolayer cells, followed by adding purified rGroEL and culturing the cells for another 48 h at 37 °C in serum-free RPMI 1640 medium. The growth of HCT-8 cells was studied using 0.1% crystal violet (CV) staining and the absorbance was measured at OD_595_. The viability of HCT-8 cells was analyzed by using a 3-(4, 5-dimethylthiazol-2-yl)-2, 5-diphenyl tetrazolium bromide assay (MTT assay), and the absorbance was detected at OD_495_. The eluent of *E. coli* BL21 (DE3) containing empty plasmid pET-26b was used instead of rGroEL as a control. Then, 5 μg/mL rGroEL was treated using 2 μg/mL anti-GroEL monoclonal antibody at 4 °C for 2 h before stimulating the host cells (cure group).

### 2.7. Detection of Cytokines

The pro-inflammatory cytokines were analyzed using an enzyme linked immunosorbent assay. Briefly, HCT-8 cells were plated on 6-well plates to yield monolayer cells, followed by adding purified rGroEL (3 μg/mL, final concentration), culturing the cells for 24 h, and identifying the cytokines in the supernatant. To verify whether the specific release of cytokines was due to rGroEL bioactivity, rGroEL pretreated with 1 μg/mL anti-GroEL monoclonal antibody for 2 h at 4 °C was used as a negative control to stimulate the host cells (anti-GroEL group). The eluent of *E. coli* BL21 (DE3) containing empty plasmid pET-26b was also used as a negative control (ck group). Tumor necrosis factor-α (TNF-α) was measured using ELISA kits (Jiancheng, Nanjing, China). Interleukin-6 (IL-6) and interleukin-8 (IL-8) assays were undertaken by the Beijing Sinouk Institute of Biological Technology.

### 2.8. Signaling Pathway Analysis

Cell culture followed the steps in Section 2.7. The culture medium was discarded and the remnants were washed three times with a PBS buffer, following which a weak RIPA buffer (Sigma) was used to digest them. This was followed by centrifugation at 4 °C (12,000× *g*, 15 min) to remove cell debris, and then, phosphatase Inhibitor (Merck, Darmstadt, Germany) and PMSF (Sigma, St Louis, MO, USA) were added to avoid protein degradation. The detection of signaling pathway proteins was carried out using a Western immunoblot. The monoclonal antibodies including anti-NF-Κb, anti-IκBα, anti-p-IκBα (Abcam, Cambridge, UK) and anti-beta actin (preserved in our laboratory) were used in this process, respectively. The specific method followed the steps in Section 2.3. As for the difference in the tight-junction protein between the GroEL-treated group and the control group, the total RNA of each group was extracted, followed by the construction of cDNA using the PrimeScript™ II Reverse Transcriptase kit (Takara, Kyoto, Japan). Subsequently, real-time quantitative PCR (RT-qPCR) was used to detect the transcription levels of the relative genes. The transcription level of the relative genes was analyzed using the 2^−ΔΔCt^ method. 2^−ΔΔCt^ method (please delete it). 

### 2.9. C. sakazakii grol Gene Silencing

The detailed protocol for inhibiting *grol* gene expression was performed as previously reported with some modifications [29]. Briefly, the protospacer adjacent motif (PAM) was designed as ‘GGACACGCCGTCTTTGGTGA’ to target the non-template DNA strand of the *C. sakazakii grol* gene. The PAM motif was linked to the pTargetF plasmid (spectinomycin resistant) with the primers F3 and R3 by reverse PCR (Appendix A). The plasmid of the pdCas9-bacteria (chloramphenicol resistant) donated inactive cas9 protein. In the case of adding spectinomycin and chloramphenicol at the same time, the bacteria containing the two aforementioned plasmids were cultured in LB medium with (CRISPRi-treated group) or without (untreated group, to remove the interference of spectinomycin and chloramphenicol) 1 μM anhydrotetracycline.

### 2.10. Bacterial Viability Assay

The MTT assay method was used to measure the bacterial viability [30]. Briefly, the bacteria were cultured to an OD_600_ of 0.6, followed by collection of the bacteria, three washes with PBS buffer, and dilution of the bacteria with fresh LB medium to an OD_600_ of 0.1. Subsequently, 200 μL bacterial suspension and 20 μL MTT (the final concentration was 0.5%) were transferred precisely in a 37 °C preheated 1.5 mL tube, followed by manually mixing the tube for a few seconds to initiate the reduction reaction and then incubating the mixture at 37 °C for 20 min to produce formazan crystals. Finally, the crystals were dissolved in dimethyl sulfoxide (DMSO), and the absorbance was measured at OD_550_.

### 2.11. Bacterial Adhesion and Invasion

Briefly, HCT-8 cells were cultured as described for the experiment to evaluate GroEL virulent effects. Logarithmic-phase bacteria were washed three times with PBS buffer to remove excess antibiotics. Next, equal amounts of different bacteria (untreated group, CRISPRi-treated group) were added into 96-well plates and incubated for 30 min. Host cells were washed three times with PBS buffer and treated with Triton X-100, followed by counting of the adherent bacteria. As for the bacterial invasion, the bacteria and HCT-8 cells were co-incubated for 90 min, followed by using 100 μg/mL gentamicin to destroy the bacteria adhering to the surface of host cells and then three washes with PBS buffer. Finally, the HCT-8 cells were treated with Triton X-100 to release the invaded bacteria and then the number of bacteria was determined.

### 2.12. Biofilm Formation Experiment

The biofilm formation experiment was analyzed through fluorescence staining. Logarithmic-phase bacteria (100 μL) were transferred to a coverglass-bottom dish and cultured statically for 3 days at 37 °C to establish a biofilm, followed by fixing the biofilm with 4% glutaraldehyde overnight at 4 °C. After removing the supernatant and washing three times with PBS, the biofilm was stained using SYBR Green I at room temperature for 30 min in the dark, followed by removing the fluorescence dye and washing three times with PBS. Finally, the morphology of the biofilm was observed under a confocal laser scanning microscope (Leica TCS SP8, Weztlar, Germany).

### 2.13. Bacterial Motility and Morphology

Bacterial motility was observed on an LB medium (0.3% agar powder). Briefly, different groups of logarithmic-phase bacteria (untreated group, CRISPRi-treated group) were added to semi-solid LB medium in an equal amount. After culturing at 25 °C or 37 °C for 12 h, the bacterial motility was observed. The morphological characteristics of the bacteria were investigated using a scanning electron microscope (SEM). Before measurement, the samples were coated with a gold layer, followed by observation of the samples in an SEM (SU1510, Hitachi, Tokyo, Japan) operating at 4700× magnification.

### 2.14. Statistical Analysis

The SPSS 18.0 software was used for statistical analysis of the data. The significant differences of the results were assessed using the unpaired *t*-test or Duncan’s test. A threshold below *p*-values of 0.05 was considered statistically significant (N * *p* < 0.05, ** *p* < 0.01, *** *p* < 0.001, **** *p* < 0.0001). At least three independent replicates were conducted for each experiment, and the results were expressed as mean ± deviation.

## 3. Results

### 3.1. Subcellular Localization and Quaternary Structure of GroEL

GroEL is a mainly cytosol-localized protein of *C. sakazakii* with an important role in the regulation of biological process of the bacteria and may contribute to the interaction of *C. sakazakii* with the host when the protein occurs extracellularly. In order to further understand the virulent roles of GroEL, the subcellular localization of this protein was investigated in detail. The localizations of GroEL in different fractions of *C. sakazakii* were confirmed based on the Western immunoblot using a GroEL-specific monoclonal antibody. As shown in Figure 1A,B, GroEL was found in all bacterial fractions, and most of it was retained in the cytoplasm. Some GroEL was also found in the periplasm and the ultracentrifuged supernatant (the fraction that removed the OMV). In the insoluble fractions (cytoplasmic membrane, outer membrane and OMV), the highest amount of the protein was detected in the OMV fraction. In order to identify whether GroEL exists on the bacterial surface, the overexpression of recombinant GroEL was detected on whole *C. sakazakii* cells through IIF. Irradiated bacteria that could recombinantly express GroEL emitted green fluorescence (Figure 1C), and there was no fluorescence signal in the control group, indicating that the GroEL was located on the surface of the bacterium. In order to know the forms of exported GroEL, the fractions of the periplasm and OMV were separated and tested using non-reduced SDS-PAGE and detected using a Western immunoblot. As shown in Figure 2A, the abundant monomer (60 kDa) and dimer (120 kDa) of GroEL were detected in the periplasmic fraction. Meanwhile, in the OMV fraction, most of this protein was mainly in the form of a monomer, and only trace amounts of dimeric GroEL (120 kDa) were detected, indicating that *C. sakazakii* exported GroEL protein mainly in the monomeric form.

### 3.2. Virulent Effects of rGroEL Protein on HCT-8 Cells

In order to verify the exact virulent roles of GroEL, the recombinant expressed protein (rGroEL) was prepared, and the virulence was investigated. After expressing recombinant GroEL protein in *E. coli* BL21 (DE3) and purifying it through affinity chromatography, the Western immunoblot was used to identify purified rGroEL. As shown in Figure 2B, the molecular weight of the rGroEL is approximately 60 kDa. The purifed rGroEL only contained monomeric and dimeric forms, which were identified by Gel permeation chromatography. After co-incubation with host cells for 30 min, the distribution of rGroEL in HCT-8 cells was analyzed using IIF and the results are shown in Figure 3A, which was taken using a confocal microscope with 400× magnification. The whole cell surface was seen to be full of green fluorescence, indicating that *C. sakazakii* GroEL could rapidly adhere to HCT-8 cells in 30 min. The adverse effects of *C. sakazakii* GroEL on HCT-8 cells was evaluated by measuring the viability of residual HCT-8 cells after co-incubation with rGroEL in a serum-free RPMI 1640 medium. With an increase in rGroEL concentration, both the viability and the number of host cells in the culture decreased. When the rGroEL concentration reached 10 μg/mL, the viability and the number of host cells reduced to 3.46% (*p* < 0.001; Figure 3B) and 4.71% (*p* < 0.0001; Figure 3C) relative to the control group, respectively. As for the cure group, the viability and the number of host cells recovered 33.82% (*p* < 0.001) and 21.03% (*p* < 0.001) compared with the group that was treated with 5 μg/mL rGroEL, respectively. These data indicate that *C. sakazakii* GroEL had a harmful effect on host cells and could induce dose-dependent apoptosis or cell necrosis, especially at a high dose.

### 3.3. Nuclear Factor Kappa-B (NF-κB) Activization and Downregulation of Tight-Junction Proteins in HCT-8 Cells under rGroEL Stimulation

To explore whether *C. sakazakii* GroEL has ability to activate intracellular signal transduction in HCT-8 cells, the expression of related proteins of NF-κB pathways was detected using a Western immunoblot. As shown in Figure 4A, there was no significant difference in the total expression of NF-κB inhibitor alpha (IκBα) under the addition of rGroEL protein, while rGroEL could promote the expression of phospho-IκBα (p-IκBα), which was an activated hallmark of NF-κB. In addition, rGroEL stimulated a concomitant increase in NF-κB, indicating that *C. sakazakii* GroEL could activate NF-κB pathways to regulate the release of pro-inflammatory cytokines. In the meantime, the relative gene transcription of tight-junction proteins were significantly reduced in rGroEL-treated cells. The quantities of gene transcription were detected through RT-qPCR, and the relative primers are shown in Appendix A. As shown in Figure 4B, the gene transcriptional levels including those of claudin-1(CLDN-1), occluding (OCLN), ZO-1 and ZO-2 decreased by 46.59% (*p* < 0.01), 23.61% (*p* < 0.05), 33.28% (*p* < 0.05), 35.62% (*p* < 0.01), respectively, which indicated that *C. sakazakii* GroEL was able to assist the bacteria to invade host cells easily.

### 3.4. Cytokine Release from HCT-8 Cells under rGroEL Stimulation

The mechanism by which rGroEL induces cell apoptosis or cell necrosis was further investigated by using an immunoenzymatic method to measure the pro-inflammatory cytokines, including TNF-α, IL-6 and IL-8, in the cell supernatant. As shown in Table 1, 3 μg/mL rGroEL could stimulate HCT-8 cells to produce 439.35 ± 14.71 pg/mL TNF-α, which was 33.68% higher than that of the control group (*p* < 0.0001). Meanwhile, 13.29 ± 0.57 pg/mL IL-6 and 11.23 ± 0.64 pg/mL IL-8 were released, which were 25.26% (*p* < 0.0001) and 110.69% (*p* < 0.001) higher than those of the control group, respectively. However, the production of the three cytokines could not be significantly (*p* > 0.05) improved when the HCT-8 cells were stimulated with rGroEL treated with a specific anti-GroEL monoclonal antibody.

### 3.5. CRISPRi-Mediated grol Gene Silencing and Bacterial Viability

In order to further examine the virulent roles of GroEL, the *grol* gene was silenced in *C. sakazakii* using the CRISPRi method. Expression of the GroEL protein was detected using SDS-PAGE. As shown in Figure 2C,D, the CRISPRi-treated group versus the untreated group, 80% downregulation in the GroEL protein expression was observed based on the analysis of the Western blot gray intensity. In order to add the same amount of viable bacteria in the following experiments, we measured the bacterial viability using the MTT assay after CRISPRi treatment. As shown in Appendix A, the bacterial viability of the CRISPRi-treated group was 327.35 ± 19.34 MRU/mL OD_600_, and there was no significant difference in viability among all the groups in the logarithmic growth phase, suggesting that *grol* gene silencing did not dramatically decrease the bacterial viability and that adding an equal amount of bacteria in the subsequent experiments could ensure the same viability for different bacterial groups.

### 3.6. Effects of GroEL on C. sakazakii Adhesion and Invasion

The foregoing adhesion-based analysis suggested that GroEL could interact with HCT-8 cells. Whether GroEL is an important participant in the interaction with HCT-8 cells in active *C. sakazakii* cells was determined by comparing the adhesion or invasion of the *grol* gene knockout strain with the control groups. As shown in Figure 5A, only 2.3 × 10^3^ CFU/well of CRISPRi-treated bacteria could adhere to host cells, accounting for 31.84% (*p* < 0.001) of the untreated group. In terms of bacterial invasion (Figure 5B), 4.18 × 10^6^ CFU/well of bacteria invaded the HCT-8 cells, covering 57.1% (*p* < 0.001) of the untreated group.

### 3.7. Effects of GroEL on Biofilm Formation, Motility and Morphology of C. sakazakii

The biofilm formation of *C. sakazakii* was evaluated through SYBR Green I staining. The results are shown in Figure 6A, the biofilm showed green fluorescence under 3D confocal scanning microscopy. The biofilm in the CRISPR-treated group showed a scattered form and faint green fluorescence, in contrast to a bright and compact biofilm in the untreated group. Through grayscale analysis, the fluorescence of the CRISPRi-treated group was approximately 57.14% that of the untreated group (*p* < 0.001), indicating the defective biofilm forming capacity of the CRISPRi-treated group. The effect of *grol* gene silencing on bacterial motility was investigated by adding an equal amount of different bacteria to semi-solid LB medium and culturing for 12 h. As shown in Figure 6B, the CRISPRi-treated group exhibited defective motility at 37 °C. As for bacterial morphology, the SEM images of bacteria were taken from the visual field at 4700× magnification (Figure 6C). Compared with the control group, the CRISPRi-treated group showed filamentation, with the number of abnormal bacteria accounting for about 3–5% and the length of the filamentous cell reaching about 10–40 μm.

## 4. Discussion

*C. sakazakii* is a redefined pathogen, and its virulence determinants are not well understood, although several common virulence factors studied in Gram-negative bacteria have been shown to affect the pathogenicity of *C. sakazakii* [8,9,10,11,12,13,14]. In this study, we elucidated that the GroEL protein was a potential virulence factor of *C. sakazakii* from three aspects including the secretion of GroEL, its virulence mechanism and its effects on bacterial character.

Most research has focused on the chaperone activity of GroEL in recent years, while in this study, the protein was found have the capability to overcome terrible obstacles composed by the *C. sakazakii* envelope and may act as a potent virulence factor in disease initiation. It is necessary that most GroEL is retained in the cytoplasm because the bacteria require the chaperone activity of GroEL to maintain protein homeostasis in the cytoplasm. Other fractions associated with GroEL indicate the complicated process of secretion of this protein. However, GroEL was mainly exported as a monomer, although trace amounts of dimeric GroEL (120 kDa) were detected in the OMV fraction (Figure 2 and Appendix A). Zhao has reported that GroEL secretion requires an intact dimeric protein complex in *Bacillus subtilis* [31]; however, we consider that the dimeric GroEL should stem from slow reconstitution of the monomer because the GroEL sequence contains the information for its folding, assembly, and function [32]. Furthermore, *C. sakazakii’s* exported monomeric GroEL appears to be more energy efficient and more flexible to recognize host receptors than the bulky dimer. The GroEL in the ultracentrifuged supernatant fraction had the same molecular weights as that in the OMV fraction, including 50, 60 and 70 kDa, while the 50 kDa GroEL was absent in the outer membrane fraction. We speculate that the 70 kDa band may contain an unknown 10 kDa protein to assist GroEL transmembrane transport because GroEL does not carry the classical signal peptide and anchor sequence. The 50 kDa band may be a degradation product. It has been reported that a GroEL nanocage could bind to the cytoplasmic membrane protein SecA, which is the core protein of the Sec system, and participate in transmembrane transport of secretory proteins by promoting secA release from the membrane in *E. coli* [33]. We consider that the misfolded GroEL might be unfolded by the GroEL–GroES nanocage and captured by SecA and subsequently secreted by the Sec system. The exported GroEL was able to exist on the bacterial surface and thus strengthen the colonization intensity of *C. sakazakii.*

To date, the toxicity of GroEL in prokaryotic organisms has been studied preliminarily only in several bacteria that cause specific tissue diseases, such as *Porphyromonas gingivalis*, in which GroEL could aggravate the alveolar inflammation and bone loss of rats, leading to the occurrence of periodontal disease [34], and *Mycobacterium tuberculosis*, in which the *grol* gene mutant could not produce granulomatous inflammation in animal experiments [35]. To our knowledge, no reports are available regarding the GroEL virulent effects of *C. sakazakii*. In this study, the *C. sakazakii* GroEL was proved to have dose-dependent virulent effects on HCT-8 cells. Furthermore, *C. sakazakii* GroEL could activate the NF-κB signaling pathway to produce more pro-inflammatory cytokines, including TNF-α, IL-6 and IL-8, which could promote a serious inflammatory reaction and hamper intestinal mucosal barrier function. Previous studies have shown that the levels of IL-6 and IL-8 were significantly increased in intestinal epithelial cells after barrier disruption [36,37]. The increased release of these cytokines, especially pleiotropic cytokine TNF-α with the potential to induce apoptosis, was closely related to the expression and the cellular redistribution of epithelial junctional proteins [38], facilitating myosin light-chain kinase (MLCK)–mediated opening of the epithelial barrier [39,40]. Here, the stimulation of *C. sakazakii* GroEL also caused the low expression of some tight-junction proteins including claudin-1, occluding, ZO-1 and ZO-2 in HCT-8 cells. The cell barrier became permeable with the reduction in the tight-junction proteins, especially at inflammatory sites [41]. These reports indicate that *C. sakazakii* GroEL may induce the dysfunction of intestinal epithelial cells by stimulating host cells to produce excessive pro-inflammatory cytokines, leading to the occurrence of bacterial translocation and inflammatory bowel diseases [40].

The GroEL–GroES system could capture more than 250 substrate proteins, and most of these proteins are involved in bacterial survival or pathogenicity [42]. When the expression of GroEL was inhibited to about 80%, *C. sakazakii* vitality was not affected, and only a slower growth of bacteria was observed (Appendix A), indicating a small amount of GroEL formed nanocages to sustain the survival of the bacteria. Biofilms have been shown to be implicated in the infection and environmental persistence of *C. sakazakii* [1], and defective biofilm formation was also found in the CRISPRi-treated group in this study. The protein kinase PrkC regulates the phosphorylation of GroEL, which contributes to the formation of GroEL-GroES nanocage, and then the GroEL nanocage mediates the folding of relevant phosphokinases, thereby promoting bacteria to form more abundant biofilms [43,44]. This beneficial cycle is damaged by *grol* gene silencing, which impairs the formation of biofilms in *C. sakazakii*. The CRISPRi-treated group also showed defective motility despite no involvement of GroEL in the folding of flagellin [42]. However, there was an abundant inclusion body of DnaK/DnaJ protein on the low expression of GroEL in *E. coli*. The DnaK/DnaJ/GrpE system held an extensive substrate network, and studies have shown that the DnaK mutant of *Clostridium difficile* lacks flagella and motility [45,46]. Therefore, the decreased motility in the CRISPRi-treated *C. sakazakii* may be caused by the impaired DnaK/DnaJ/GrpE system. The *grol* gene silencing also aggravated *C. sakazakii* filamentation, which may be attributed to the cell-division proteins FtsE [47]. Additionally, the bulkily filamentous cells will further damage the flexibility of the bacteria and thus impair the motility of *C. sakazakii*. However, the production of LPS and lipid A were not affected (Appendix A) despite the involvement of some substrate proteins of GroEL in carbohydrate/lipid transport and metabolism [39]. In short, the abovementioned physiological defects caused by deficient GroEL will weaken the pathogenicity of *C. sakazakii*.

## 5. Conclusions

In conclusion, this study proved that the GroEL protein is a potential virulence factor. *C. sakazakii* could mainly export monomeric GroEL protein through its OMVs. The exported GroEL could exist on the surface of the bacteria and was able to quickly adhere to human enterocyte-like epithelial cells, indicating that GroEL contributed to bacterial adhesion and colonization. GroEL could individually activate the NF-κB signaling pathway to induce the inflammatory response and downregulated the expression of some tight-junction proteins, ultimately causing necrosis of the host cell, which would be beneficial for the bacteria to cross the intestinal barrier. In addition, other properties associated with the pathogenicity of the bacteria, including biofilm formation, motility and morphological character, were controlled by the expression of GroEL. These abovementioned facts indicate that GroEL is important for bacterial pathogenicity.

## Figures and Tables

**Figure 1 foods-12-03404-f001:**
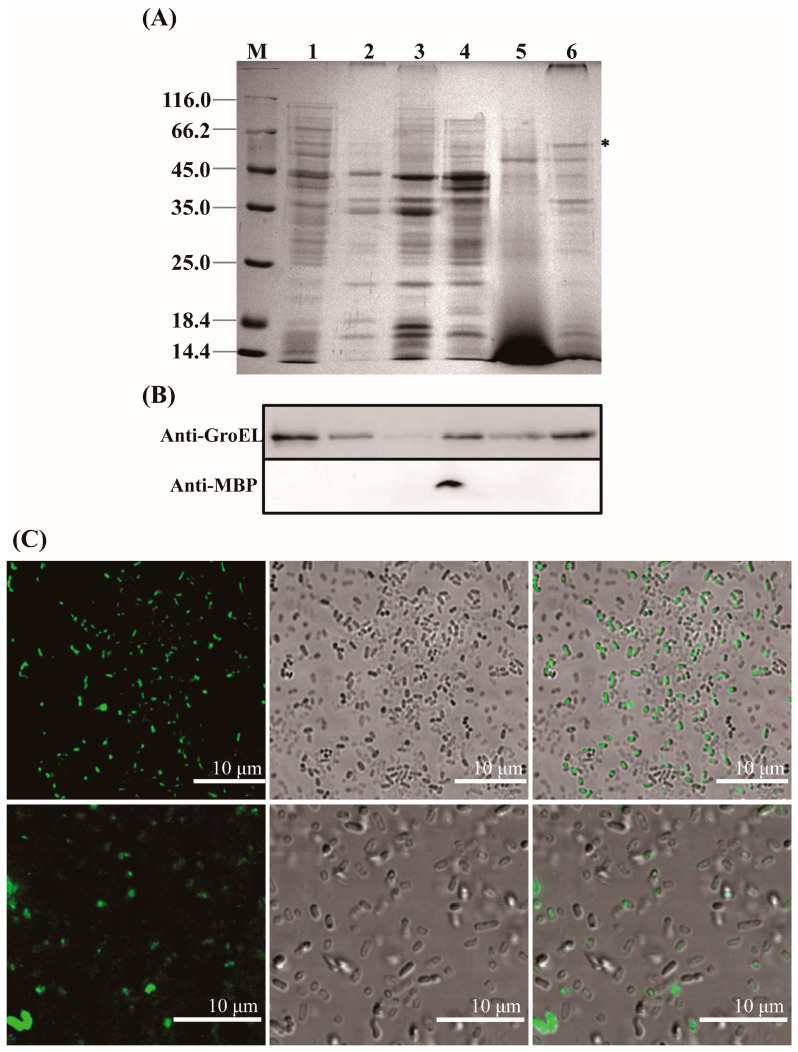
Localization of the GroEL protein in *C. sakazakii*. (**A**) SDS-PAGE of different fractions of *C. sakazakii* and the band of GroEL (60 kDa) is labeled with an asterisk; each lane contains 10 μg proteins except for ultracentrifuged supernatant fraction (30 μg). M, protein marker; Lane 1, cytoplasm; Lane 2, cytoplasmic membrane; Lane 3, outer membrane; Lane 4, periplasm; Lane 5, ultracentrifuged supernatant; Lane 6, OMV. (**B**) Identification of GroEL (top) and the periplasmic maltose binding protein MBP (dwon) in various bacterial fractions using Western immunoblot, the band of GroEL (60 kDa) is labeled with an asterisk (please delete it). MBP (43.4 kDa) was used to evaluate bacterial lysis, and anti-MBP monoclonal antibody (Abclonal, Wuhan, China) was used to identify this protein. Lane 1, cytoplasm (10 μg); Lane 2, cytoplasmic membrane (10 μg); Lane 3, outer membrane (10 μg); Lane 4, periplasm (10 μg); Lane 5, ultracentrifuged supernatant (30 μg); Lane 6, OMV (10 μg). (**C**) Indirect immunofluorescence detection of GroEL protein associated with bacterial surface. The fluorescence signal was taken from the visual field at 400× (top) and 630× magnification (bottom), respectively. * *p* < 0.05.

**Figure 2 foods-12-03404-f002:**
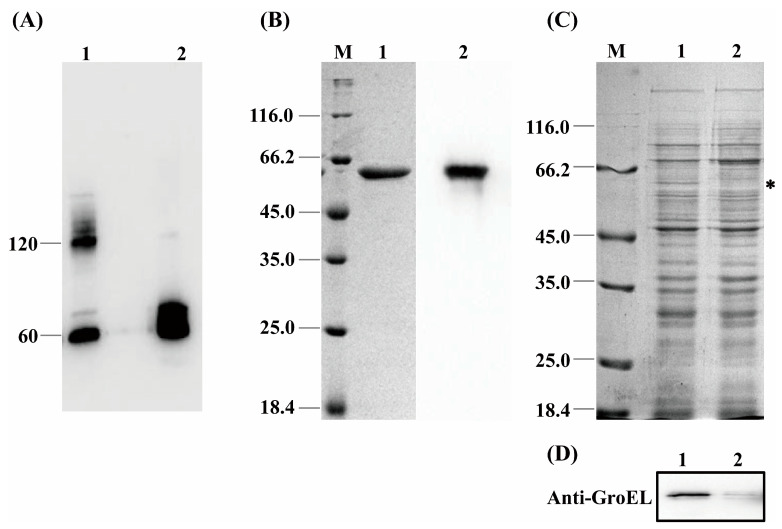
(**A**) The quaternary structure of GroEL in periplasmic fraction and OMV. Lane 1, periplasm (40 μg); Lane 2, OMV (60 μg). (**B**) Purification of recombinant GroEL protein through affinity chromatography. M, protein marker; Lane 1, SDS-PAGE profile depicting purification of rGroEL that has removed the signal peptide of plasmid vector; Lane 2, identification of rGroEL using Western immunoblot. (**C**) SDS-PAGE profile analyzing the expression level of GroEL protein of bacteria, and the band of GroEL (60 kDa) is labeled with an asterisk. M, Marker; Lane 1, untreated group (10 μg); Lane 2, CRISPRi-treated group (10 μg). (**D**) Quantification of relative expression of *grol* gene using Western immunoblot. Lane 1, untreated group (10 μg); Lane 2, CRISPRi-treated group (10 μg). * *p* < 0.05.

**Figure 3 foods-12-03404-f003:**
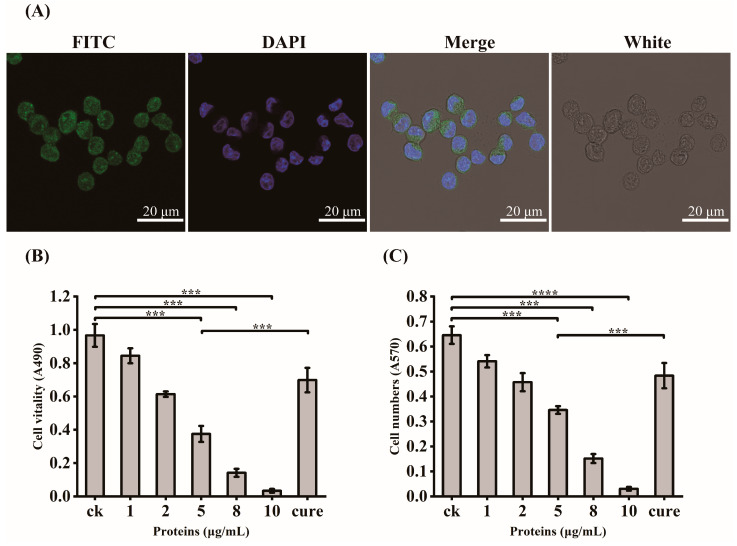
GroEL virulent effect determination. (**A**) IIF for the localization of recombinant GroEL in HCT-8 cells, 400× magnification; FITC, identification of recombinant GroEL on the surface of HCT-8 cells using FITC fluorescence staining; DAPI, fluorescent staining of cell nuclei by DAPI; White, observation of HCT-8 cells at light microscope, Merge, the overlaying of the above three images. (**B**) changes in the vitality of HCT-8 cells after GroEL treatment; (**C**) changes in the number of HCT-8 cells after GroEL treatment. Three independent replicates were conducted for each experiment. *** *p* < 0.001, **** *p* < 0.0001.

**Figure 4 foods-12-03404-f004:**
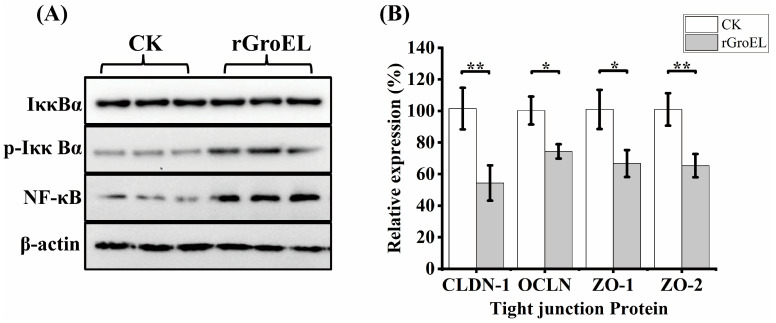
GroEL contributes to NF-κB activation and intestinal cell permeability. (**A**) Immunoblot of GroEL-induced relative signaling proteins of NF-κB pathways levels in HCT-8 cells. β-Actin was used as control. (**B**) RT-qPCR detected the transcription level of related tight-junction protein genes of HCT-8 cells. The gene glyceraldehyde-3-phosphate dehydrogenase (GAPDH) was used as control. And each experiment was repeated at least three times. * *p* < 0.05, ** *p* < 0.01.

**Figure 5 foods-12-03404-f005:**
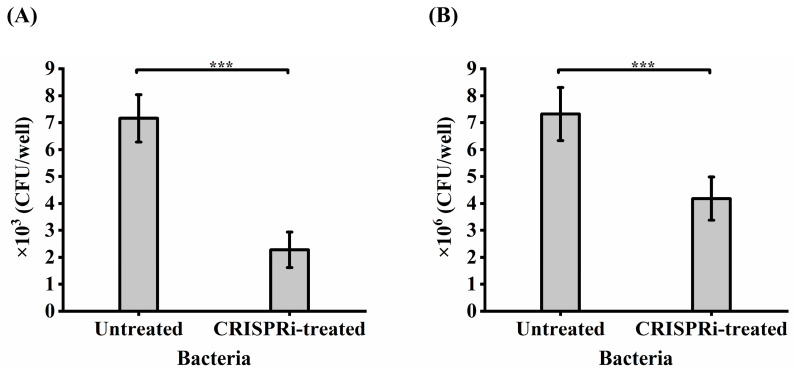
Adhesion (**A**) and invasion (**B**) of bacteria to HCT-8 cells. Each experiment was repeated three times. *** *p* < 0.001.

**Figure 6 foods-12-03404-f006:**
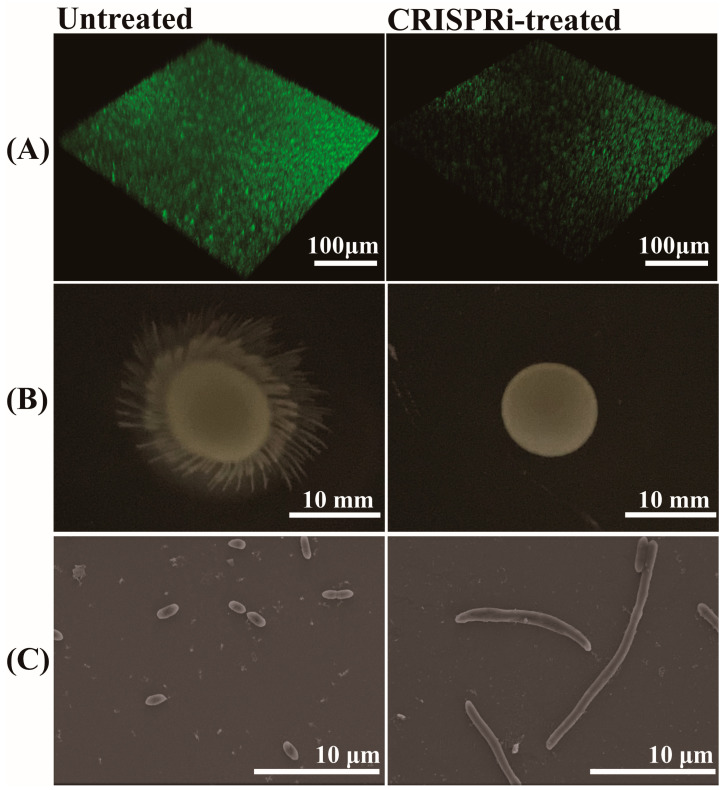
(**A**) Quantification of biofilm via SYBR Green I stain; (**B**) bacterial motility assay at 37 °C; (**C**) bacterial morphological observation through SEM.

**Table 1 foods-12-03404-t001:** Determination of pro-inflammatory factors released by HCT-8 cells under the stimulation of GroEL.

Pro-Inflammatory Cytokines (pg/mL)	Control	rGroEL	rGroEL + Anti-GroEL
TNF-α	328.66 ± 21.36	439.35 ± 14.71	344.97 ± 28.47
IL-6	10.61 ± 0.38	13.29 ± 0.57	10.89 ± 1.32
IL-8	5.33 ± 0.55	11.23 ± 0.64	6.23 ± 1.43

## Data Availability

The data used to support the findings of this study can be made available by the corresponding author upon request.

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
