# Peer review of "Characterization of Molecular Chaperone GroEL as a Potential Virulence Factor in Cronobacter sakazakii"

_foods, 2023, doi:10.3390/foods12183404_

Round 1
Reviewer 1 Report
Zhu et al. in this study proved that C. sakazakii exported GroEL mainly in monomeric form and they showed that this protein adheres to human enterocyte-like epithelial cells subsequently activate NF-κB signaling inducing the inflammatory response, and downregulates some tight junction proteins which assist bacteria to cross intestinal barrier. They also performed additional experiments and showed that GroEL protein is involved in biofilm formation, motility and bacterial filamentation. This study is interesting and may add some knowledge to the existing field. Authors have to address some of the important comments to improve the quality of the manuscript.
- In the abstract, it is suggested to introduce the importance of GroEl in food science.
- Methods for statistical tests were missing
- Statistical information in figure legends and a number of replicates used in experiments were not mentioned.
- In figure 1 A and 2 C: Western blot images show many bands like in Coomassie Brilliant Blue staining—What is the specificity of the antibodies used for Western blotting? A couple of crossreactive proteins may be detected with polyclonal antibodies, but in this case, it looks like a Coomassie-stained bands.
- What are those two insets below the Figure 1A and Figure 2C. Details were not mentioned in the figure legend. Are those Western blots of purified fractions separated from the full image of Western blot?
Reviewer 2 Report
Dear authors
Since the idea and information provided of this current paper titled “Characterization of molecular chaperone GroEL as a potential virulence factor in Cronobacter sakazakii » are interesting. But, some points which should be addressed in order to improve the quality of the MS.
Abstract section
1. Some technical approaches used in this investigation should be developed briefly
2. A 2-3 concise and conclusive sentences should be added at the end of the abstract
- Introduction section
3. This sections should be shorten
4. I invite authors to use recent and proper references (2019-2023), and more sentences should be developed
5. Some clinical data linked to Cronobacter sakazakii should be introduced
6. The thermal tolerance of Cronobacter sakazakii in food industry should be discussed
7. L 47, Gram
8. L61-63, please develop this sentence
9. How about the practical applicability of this study on the food industry field?
10. The authors should stress the novelty of this work
11. The objective was not clear, improve it
Material and methods section
12. L99-100 authors shoud enumerate the exact number of Cronobacter sakazakii
13. This sub section “2.2. Bacterial fractionation 97 should be concise
14. Please add a suitable reference in the subsection 2.3;
15. L178, the last sentence should be in statistical analysis at the end of MM section
16. Some abbreviation as MTT should be indicated, I recommend to add a subsection of abbreviation list after the abstract section
17. MTT at MTT (5.0 g/L of what?)
18. A complete statistical analysis should be added at the end of MM section
19. All data should well linked since authors have several responses and data
Results section
20. All tables and figures should be inserted in the main text according to mdpi journal’s instructions
21. L270-271, please be clear and develop more this sentence
22. In this section, authors should focus on the main and original results
23. If authors can ‘transform the figure 5 on Table?
Discussion section
24. all data should be linked, since this study has several experiments
25. authors should deeply discuss their results, and compare their results with another recent and suitable works
The conclusion part
26. should be improved taking into all remarks and suggestions
